# PD-L1 Expression in Endometrial Cancer and Its Association with Clinicopathological Features: A Systematic Review and Meta-Analysis

**DOI:** 10.3390/cancers14163911

**Published:** 2022-08-13

**Authors:** Mohd Nazzary Mamat @ Yusof, Kah Teik Chew, Nirmala Kampan, Nor Haslinda Abd. Aziz, Reena Rahayu Md Zin, Geok Chin Tan, Mohamad Nasir Shafiee

**Affiliations:** 1Gynaecologic-Oncology Unit, Department of Obstetrics and Gynaecology, Universiti Kebangsaan Malaysia Medical Centre, Kuala Lumpur 56000, Malaysia; 2Department of Pathology, Universiti Kebangsaan Malaysia Medical Centre, Kuala Lumpur 56000, Malaysia

**Keywords:** endometrial cancer, PD-L1, immune checkpoint, clinicopathological, prevalence

## Abstract

**Simple Summary:**

In women, endometrial cancer is a crucial cancer cause-death, which is still not fully explored in its pathogenesis and immune system. Early detection is essential for proper treatment and follow-up in affected patients. This systematic review and meta-analysis aim to pool the prevalence of PD-L1 in endometrial cancer and its association with clinicopathological features. The pooled prevalence of PD-L1 was 34.26% in tumour cells, and 51.39% in immune cells among endometrial cancer patients. There was significant association of PD-L1 expression in both tumour cells and immune cells with advanced stage endometrial cancer. The presence of lympho-vascular invasion and poor overall survival were also associated with PD-L1 expression in immune cells. These information enable clinicians to stratify endometrial cancer patients for anti-PD-1/PD-L1 immune therapy.

**Abstract:**

Endometrial cancer (EC) is one of the most common malignancies of the female genital tract and its current treatment mainly relies on surgical removal of the tumour bulk, followed by adjuvant radiotherapy with or without chemotherapy/hormonal therapy. However, the outcomes of these approaches are often unsatisfactory and are associated with severe toxicity and a higher recurrence rate of the disease. Thus, more clinical research exploring novel medical intervention is needed. Involvement of the immune pathway in cancer has become important and the finding of a high positive expression of programmed cell death-ligand 1 (PD-L1) in EC may offer a better targeted therapeutic approach. Numerous studies on the PD-L1 role in EC have been conducted, but the results remained inconclusive. Hence, this systematic review was conducted to provide an update and robust analysis in order to determine the pooled prevalence of PD-L1 expression in EC and evaluate its association with clinicopathological features in different focuses of tumour cells (TC) and immune cells (IC). A comprehensive literature search was conducted using the PubMed, Web of Science, and Scopus databases. Twelve articles between 2016 and 2021 with 3023 EC cases met the inclusion criteria. The effect of PD-L1 expression on the outcome parameters was estimated by the odds ratios (ORs) with 95% confidence intervals (CIs) for each study. The pooled prevalence of PD-L1 was 34.26% and 51.39% in the tumour cell and immune cell, respectively, among women with EC. The PD-L1 expression was significantly associated with Stage III/IV disease (in both TC and IC) and correlated to the presence of lympho-vascular invasion in IC. However, the PD-L1 expression in TC was not associated with the age groups, histology types, myometrial invasion, and lympho-vascular invasion. In IC, PD-L1 expression was not associated with age group, histology type, and myometrial invasion. The meta-analysis survival outcomes of PD-L1 high expression had a significant association with worse OS in IC but not in TC.

## 1. Introduction

Endometrial cancer (EC) is a heterogeneous group of tumours derived from endometrial glandular epithelium with several different histologic types differing in their morphologic and molecular features. There is also intratumoural morphologic heterogeneity, with different neoplastic cell components within the same tumour, with different morphologic and molecular features [1,2]. According to Global Cancer Statistics 2020, EC was the sixth most common diagnosed cancer among women worldwide [3]. It is estimated about 65,950 new cases of EC with 12,550 deaths will occur in 2022 [4]. More than 90% of endometrial cancer occurred in women over the age of 50 years old [5]. Most of the cases (80%) detected were in the early stages with good prognosis (95% 5-year survival rate) [6]. Prognostic factor evaluation is crucial for the clinical management and treatment regimen for the determination of EC. The previous literature has suggested that several factors such as obesity, diabetes, and insulin resistance are linked with poor prognosis in EC. However, predicting the survival of an individual patient with EC remains difficult and challenging [7]. Therefore, a reliable predictive marker is urgently needed to monitor the disease progression of EC.

Cancer progression is often accompanied by various mutations, which provide cancer cells with a selective advantage by increasing their genetic diversity and thus accelerating their evolutionary fitness. However, this diversity comes with a price: the more a cancer cell differs from a normal cell, the more likely it is to be mistaken as a foreign object by the immune system [8,9,10]. Previously, the role of the immune system in cancers was overlooked because tumour cells effectively suppress immune responses by adopting features that enable them to escape immune system detection [9,10,11]. In recent decades, several inhibitory immunoreceptors including programmed cell death protein 1 (PD-1) and its ligand (PD-L1), cytotoxic T-lymphocyte protein 4 (CTLA-4), lymphocyte-activation gene 3 (LAG3), T cell immunoglobulin and mucin-domain containing-3 (TIM3), T cell immunoreceptor with Ig and ITIM domains (TIGIT), and B- and T-lymphocyte attenuator (BTLA) have been identified and evaluated in various cancers. Immune checkpoints are molecules that operate as gatekeepers of immune responses [10]. The two most garnered attention immune checkpoints are PD-1 and CTLA-4. CTLA-4 is a negative regulator of T cells that acts to control T-cell activation by competing with the co-stimulatory molecule CD28 or binding to shared ligands CD80 (also known as B7.1) and CD86 (also known as B7.2) [9,10,12].

The PD-1 axis is an immunosuppressive pathway that allows tumour cells to remain undetected by the human immune system [13]. PD-L1 positive expression has been proven to be a good prognostic marker in various cancers including colorectal cancer, pancreatic cancer, oesophageal squamous cell carcinoma, and lung cancer [14,15]. The literature reported that in squamous cell lung carcinoma, increased PD-L1 expression was linked to a shorter lifespan [16]. PD-L1 has been studied for its role in EC with inconclusive results [17,18,19] whereby several studies have found that PD-L1 expression does not have a substantial prognostic effect in EC [17]. However, Yamashita et al. (2018) found a link between increased PD-L1 expression and a better progression-free survival in EC [20]. As a result, we used the existing literature to conduct a meta-analysis of several independent investigations to pool and analyse the association of clinicopathological features with PD-L1 expression among women with EC.

## 2. Materials and Methods

### 2.1. Protocol and Eligibility Criteria

This systematic review and meta-analysis followed the standard guideline protocol provided by the Preferred Reporting Items for Systematic Reviews and Meta-Analysis (PRISMA) [21] and was registered in the PROSPERO database (CRD42021293537).

All included studies were focused on PD-L1 expression in EC and written in English. All studies also included demographic features such as age, percentage of PD-L1 expression, tumour stage, histology type, myometrial invasion, and lympho-vascular invasion. Any review paper, book, practice guideline, letter, editorial, commentary, case report, and pilot study were all excluded.

### 2.2. Information Sources and Search Criteria

The search was conducted in the Google Scholar, PubMed, and Web of Science databases. The main fraction keywords using the PICO tool was focused on the Population, Intervention, Comparison, and Outcomes [22]. The population was defined as women or patients with EC. The intervention was defined as the PD-L1 expression status in patients with EC. The comparison was defined as the association of clinicopathological features with PD-L1 expression. The outcome was defined as the prognosis and progress of EC. The search strategy used the keyword concept by medical subject heading (MeSH) terms. Three techniques of database search were employed including: quotation mark (double quote, “ ” for the terms to keep together and in order), parentheses (allow to combine concepts), and Boolean operators (AND/OR relationship). Databases were searched using the keywords of (“PD-L1” OR “programmed death ligand 1” OR “programmed cell death ligand 1” OR “B7-H1” OR “CD274”) AND (“endometrial cancer” OR “endometrial carcinoma” OR “endometrium cancer” OR “endometrial neoplasms”). The literature search was set to a ten year timeline, limited to articles from 2011 to 2021 and was conducted on 30 November 2021. Open access and restricted access articles were reviewed by two authors independently (M.N.M.@Y. and K.T.C.) and the articles were retrieved by the UKM library institutional support. The identified studies were uploaded into Mendeley, and duplicate articles were removed.

### 2.3. Study Selection

The preliminary selected articles were assessed for their validity and relevance with the inclusion criteria. The validity of individual studies was screened for information, selection bias, and the quality of data analysis. Titles and abstracts were screened for the related information and unrelated articles were eliminated. Next, the selected articles based on the eligibility criteria were analysed in full-text. Articles included for the final analysis were assessed for risk of bias. Data extraction was performed for those articles.

### 2.4. Data Collection Process and Data Items

The data were extracted into the table explicitly designed for this review, which was listed with the authors, year of publication, age, number of cases included, percentage of PD-L1 expression, tumour stage, histology type, myometrial invasion, and lympho-vascular invasion.

### 2.5. Assessment of Risk of Bias of Individual Studies

The Newcastle–Ottawa scale (NOS) of the cohort study was used to assess the quality of the studies that were eligible [23]. The NOS contains nine stars divided into three categories: selection (four stars), comparability (two stars), and outcome (three stars). An NOS score of 0 to 9 was used to indicate the quality of the studies, and studies with NOS ≥6 were considered as high quality with low publication bias.

### 2.6. Statistical Methods

The meta-analysis proportion of the quantitative data for the prevalence of TC and IC with PD-L1 expression among the endometrial cancer cases was pooled using MedCalc software (version 19.4). The pooled prevalence was reported as the 95% confidence interval (CI), and statistical heterogeneity was assessed by using the Cochrane Q statistic and I-squared (I^2^) statistics. Heterogeneity was statistically significant when the *p*-value of the Cochrane-Q test was less than 0.05 and the I^2^ statistic value was higher than 50%. When the heterogeneity I^2^ was low, the fixed effect models (FEM) were used and if substantial heterogeneity I^2^ was encountered, random or the quality effect models (QEM) were used to pool the prevalence.

A meta-analysis of associations between TC and IC of PD-L1 in EC and the clinicopathological features were performed using Review Manager (RevMan) software, Version 5.4. Copenhagen. The results with a *p*-value of less than 0.05 were of statistical significance. Heterogeneity among the studies was calculated using the chi-square (χ^2^) test and determined by the I^2^ statistic. Forest plots were drawn to visualise the evaluation of findings in individual studies and overall (global) studies for each association. The pooled HRs and 95% CIs of the individual study was used to evaluate the association between PD-L1 and OS. An HR > 1 without a 95% CI containing 1 indicated that PD-L1 overexpression predicts lower survival.

A funnel plot of the included studies presented to evaluate the risk of publication bias showed a symmetrical inverted funnel shape for the low risk of publication bias and vice versa. Publication bias evaluation also conducted using the Egger’s test. The sensitivity analysis of the results was also evaluated using the method leave-one-out.

## 3. Results

### 3.1. Search Sequence and Quality Assessment of Selected Publications

A total of 412 studies obtained through electronic search, and 229 studies were excluded due to duplication (Figure 1). The titles and abstracts of the remaining 183 records were then screened. An additional 48 studies were excluded, and 135 full-text studies were deemed to be relevant and were examined in detail. Out of these, 123 full-text articles were excluded for the reasons listed in Figure 1. The remaining 12 studies were all evaluated for risk of bias, according to NOS (Table 1). The NOS scores of 6 to 8 indicated a low risk of bias of all of the included studies.

### 3.2. Studies Characteristics

A total of 12 studies, with 3023 EC cases, were finally included in this review [17,18,19,24,25,26,27,28,29,30,31,32]. All studies used the data from immunohistochemistry (IHC) for positive PD-L1 expression. Most of the studies used a cut-off at more than 1% IHC staining as the positive expression (nine studies). Only three studies used different cut-off value where one study used a 5% cut-off value [18], one study used a 10% cut-off value [25], and another one study did not clearly state the cut-off value [32]. The earliest study was published in May 2016 by Mo and colleagues [19], and the most recent study was published in February 2021 by Siraj and colleagues [18]. The most extensive study was conducted by Pasanen et al. (2019) [27] with 842 patients, and the smallest study was carried out by Chew et al. (2020) [31] and Kir et al. (2020) [28], which included 59 patients each. All of the included studies and variables of interest associated with PD-L1 expression in EC are summarised in Table 2.

### 3.3. Quantitative Synthesis

#### 3.3.1. Prevalence of PD-L1 Expression in Endometrial Cancer

The total number of PD-L1 expression on the TCs and ICs for further analysis was detected by IHC from the tissue samples. The heterogeneity was significant for the pooled prevalence of PD-L1 expression among EC, with a *p*-value of the Cochrane Q statistics less than 0.0001 and I^2^ statistic values with 98.65% (TC) and 97.48% (IC). From 11 studies that reported PD-L1 expression on the TCs, the overall pooled prevalence among the EC cases was 34.26% [95% CI: 19.46–50.82] based on the QEM (Figure 2). Among the included studies in this review, which evaluated the expression of PD-L1 in TCs, Pasenan et al. (2019) [27] reported the lowest prevalence of PD-L1 expression (8.58%), and Zhang et al. (2020) [24] reported the highest prevalence (70.14%). On the other hand, the expression of PD-L1 in ICs, Zhang et al. (2020) [24] reported the lowest prevalence, which was only 16.29% and Sungu et al. (2019) [26] reported the highest prevalence of 72.44%. For the ICs, seven studies reported PD-L1 expression with an overall pooled prevalence of 51.39% [95% CI: 33.84–68.77%] based on the QEM (Figure 3).

#### 3.3.2. Analysis of PD-L1 Expression in Tumour Cells (TCs) of Endometrial Cancer

##### Age Group and PD-L1 Expression in TCs

The association between age group and PD-L1 expression on TCs was analysed for five out of 11 studies based on the available data. The PD-L1 expression on the tumour cells was not associated with the age groups among women with EC, as shown in the pooled analysis [OR = 0.99; 95% CI = 0.42–2.32] by using QEM, as the heterogeneity was significant [*p* = 0.0002; I^2^ = 82%] (Figure 4A).

##### Tumour Stages and PD-L1 Expression in TCs

Available data from six out of 11 studies were analysed for the association between tumour stage (Stage I/II versus Stage III/IV) and the PD-L1 expression on TCs. The FEM was used as there was no significant heterogeneity [*p* = 0.05; I^2^ = 54%]. The PD-L1 expression on the TCs was significantly linked with Stage III/IV disease [OR = 0.65; 95% CI = 0.52–0.81] (Figure 4B).

##### Histology Types and PD-L1 Expression in TCs

Analysis for the association between histology type (endometroid versus non endometroid) and PD-L1 expression on the TCs was conducted in nine studies based on the available data. The QEM was used as there was a significant heterogeneity [*p* < 0.00001; I² = 96%]. The analysis revealed that there was no significant association between the histology type and PD-L1 expression on the TCs [OR = 1.21; 95% CI = 0.30–4.91] (Figure 4C).

##### Myometrial Invasion with PD-L1 Expression in TCs

The association between myometrial invasion (less than 50% versus more and equal than 50% invasion) and PD-L1 expression on TCs was explored in six out of 11 studies based on the available data. The QEM was used as there was significant heterogeneity [*p* < 0.00001; I² = 91%]. The statistical analysis failed to demonstrate a significant association between the myometrial invasion and PD-L1 expression on the TCs [OR = 1.21; 95% CI = 0.30–4.91] (Figure 4D).

##### Lympho-Vascular Invasion with PD-L1 Expression in TCs

The analysis for the association between lympho-vascular invasion (yes versus no) and PD-L1 expression on TCs was conducted in six out of 11 studies. QEM was used as there was significant of heterogeneity [*p* < 0.00001; I² = 85%]. There was no significant association between lympho-vascular invasion and PD-L1 expression on TC [OR = 1.25; 95% CI = 0.47–3.36] (Figure 4E).

#### 3.3.3. Analysis of PD-L1 in Immune Cells (ICs) of Endometrial Cancer

##### Age Group and PD-L1 Expression in ICs

Available data from four out of seven studies were analysed for the association between the age group and PD-L1 expression on ICs. QEM was used as there was a significant heterogeneity [*p* = 0.03; I^2^ = 66%]. There was no significant association between PD-L1 expression on ICs with age group (Figure 5A).

##### Tumour Stages and PD-L1 Expression in ICs

Three studies evaluated the association between tumour stages (Stage I/II versus Stage III/IV) and PD-L1 expression on the ICs. FEM was used as there was no significant heterogeneity [*p* = 0.97; I^2^ = 0%], and the results revealed that there was a significant association between Stage III/IV and PD-L1 expression on ICs [OR = 0.66; 95% CI = 0.54–0.81] (Figure 5B).

##### Histology Type and PD-L1 Expression in ICs

The association between the histology type (endometroid versus non endometroid) and PD-L1 expression on IC was investigated in five out of seven studies. QEM was used as there was significant heterogeneity [*p* = 0.0001; I² = 83%]. There was no significant association between histology type and PD-L1 expression on ICs [OR = 0.88; 95% CI = 0.19–4.03] (Figure 5C).

##### Myometrial Invasion with PD-L1 Expression in ICs

Only two studies evaluated the association between myometrial invasion and PD-L1 expression on ICs. QEM was used as there was a significant heterogeneity [*p* = 0.02; I^2^ = 82%], and the results revealed that there was no significant association between myometrial invasion and PD-L1 expression on the ICs [OR = 0.94; 95% CI = 0.24–3.67] (Figure 5D).

##### Lympho-Vascular Invasion with PD-L1 Expression in ICs

A study on the association between lympho-vascular invasion (yes versus no) and PD-L1 expression on the ICs was conducted in five out of seven studies based on the available data of the studies. FEM was used as there was no significant heterogeneity [*p* = 0.21; I² = 32%]. There was a significant association between the presence of lympho-vascular invasion and PD-L1 expression on immune cells [OR = 2.05; 95% CI = 1.27–3.28] (Figure 5E).

#### 3.3.4. PD-L1 Expression and Survival Outcomes

A total of two studies provided data on PD-L1 expression and overall survival (OS) in TCs [17,24], and the heterogeneity was statistically significant (I^2^ = 0%, *p* = 0.55); therefore, the FEM method was used. The pooled data were as follows: HR = 0.52, 95% CI = 0.26–1.03, *p* = 0.06, indicating that PD-L1 high expression had a non-significant association with OS (Figure 6A). In the case of the ICs, two studies provided data on the PD-L1 expression and OS [24,32]. The heterogeneity was statistically significant (I^2^ = 0%, *p* = 0.39); therefore, FEM was used. The pooled data were as follows: HR = 2.04, 95% CI = 1.05–3.97, *p* = 0.05, indicating that PD-L1 high expression had a significant association with worse OS in the ICs (Figure 6B).

#### 3.3.5. Publication Bias and Sensitivity Analysis

By using Begg’s test, we estimated the publication bias of the included studies regarding to OS. As shown in Figure 7A (TC), there was no clear evidence of funnel plot asymmetry by visual assessment but for the IC (Figure 7B), there was an asymmetry funnel plot. The Begg’s *p*-values were *p* = 0.432 for TC and *p* = 0.356 for IC, which suggested that there was no significant publication bias in this meta-analysis.

In addition, the sensitivity analysis revealed that the variation range of the pooled results of PD-L1 proportion in TC was [ES (95% CI lowest range–highest range) = 0.34 (0.20, 0.45) to (0.26, 0.50), *p* < 0.05] (Figure 8A and Table 3), and in IC, [ES (95% CI lowest range–highest range) = 0.52 (0.31, 0.66) to (0.39, 0.73), *p* < 0.05] (Figure 8B and Table 3) of EC patients were not significantly reversed. Similarly, the variation range of the pooled results for other analysis, as listed in Table 3, which had a range within the ES and P < 0.05. However, two analyses showed a significant reverse: there was an association analysis of PD-L1 and OS analysis of TC [0.52 (0.06,0.89) to (−0.76, 2.28), *p* > 0.05, Table 3], and am OS analysis of IC [2.04(−2.63, 4.31) to (0.55, 3.95), *p* > 0.05, Table 3]. These results suggest that the pooled results for most of the outcomes were stable and not significantly altered by any single study, except for the analysis of the association analysis of the PD-L1 and OS survival analysis in both the TC and IC.

## 4. Discussion

PD-L1 is an important immune regulatory factor, and it plays a key role in the immune escape mechanism of cancer cells. PD-L1 specifically binds to the PD-1 receptor of T cells and affects the activation and differentiation of T cells. Tumour-infiltrating immune cells have been shown to induce cytokines such as interferons and vascular endothelial growth factors that upregulate PD-L1 expression [33,34,35]. PD-L1 upregulation is known to regulate various intracellular signalling pathways both at the transcriptional and translational levels [36,37]. Therefore, the high expression of PD-L1 may potentially influence cancer progression and be associated with poor prognoses. Studies on the expression, regulation, and function of the PD-1/PD-L1 pathway in the human cancer microenvironment have provided scientific evidence that has directly supported the current clinical application for blocking the PD-1/PD-L1 pathway. Since the PD-1/PD-L1 pathway is suggested to play a crucial role in the immune escape mechanism and growth of cancer cells, the relationship between PD-L1 expression and its role in EC has markedly attracted many researchers and clinicians [19,24,26,31]. However, previous studies have yielded discrepant results. Thus, the issue remains controversial concerning EC. Therefore, we conducted a systematic search for relevant studies on this topic, followed by a meta-analysis of the findings. As for our knowledge, this is the first meta-analysis prevalence of PD-L1 expression and its association with clinicopathological features, which focuses on TCs and ICs among women with EC.

The overall pooled prevalence of PD-L1 expression was 34.26% in TC and 51.39% in IC among women with EC. A higher PD-L1 prevalence in IC compared to TC was concordant with a previous meta-analysis on breast cancer by Boman et al. (2021) [38], which reported a PD-L1 expression of 18.7% in TC and 51.2% in IC. A similar finding was also reported in an individual study of gastro-oesophageal cancers as no meta-analysis was available, which showed that the PD-L1 expression of TC had a range of 14% to 24% and was 35% in IC [39]. Another individual study by Fakhri et al. (2021) [40] on non-small cell lung cancer also reported higher PD-L1 expression in IC (34.4%) compared to TC (27.2%). These findings support the idea that ICs not only function as a restriction of T-cell activity (by supplying a source of PD-L1), but also facilitate the initial intratumoural expansion of T cells [41,42,43]. Higher PD-L1 expression indicated that ICs play a critical role in regulating T-cell responses independently, other than PD-L1 expression by TC. Thus, this demonstrates the role of the IC expression of PD-L1 as a vital indicator of pre-existing immunity and active immune suppression in the tumour microenvironment [41].

Forest plots pooled in this meta-analysis revealed that the expression of PD-L1 was significantly associated with an advanced stage of EC (stage III/IV) in both TC and IC. The findings were consistent with the previous meta-analysis [44], and other cancers such as in colorectal cancer [45] and pancreatic cancer [46]. As the cancers progress, studies have shown that the upregulation of PD-L1 was due to the activation of the Janus kinase 2/signal transducers and activators of the transcription 1 (JAK2/STAT1) signalling pathway in EC and other cancers such as colorectal, pancreatic, and gastric cancer [47,48,49,50]. As a result, PD-L1 expression will be higher among women diagnosed at an advanced stage of EC. These women will suffer a poorer outcome with limited treatment options, resulting in a low survival rate [30,31]. These findings support the idea of an interaction between the tumour and its surrounding inflammatory microenvironment [26]. Hence, there may be a potential use of the PD-L1 immuno-checkpoint inhibitor in patients with an advanced stage as a therapeutic strategy.

This meta-analysis also found a significant association between the PD-L1 expression and lympho-vascular invasion in the ICs of women with EC, but not in TCs. Previous experimental studies have shown that there was a positive link between the PD-L1 expression and lympho-vascular invasion. Epithelial-to-mesenchymal transition (EMT) leads to lymphatic invasion, and the expression of PD-L1 in the tumour cells facilitates immunosuppression, both of which contribute to tumour progression and metastasis. As a result, lymphatic invasion and PD-L1 overexpression might be thought of as “parallel” in nature [45,51]. On the other hand, no significant differences were observed in the pooled forest plot between PD-L1 expression and age group, histology type, myometrial invasion, and in TCs. In ICs, the pooled results also showed that there was no association between PD-L1 expression and age group, histology type, and myometrial invasion.

In this meta-analysis, PD-L1 expression had no significant association with OS in TC among EC patients. However, high PD-L1 positive expression in IC was associated with worse OS. PD-L1 expressed on the surface of TC is supposed to bind to the PD-1 receptor on immune cells and to induce adaptive immune resistance [24]. Our observation may be explainable if some proportion of expressed PD-L1 could move between the surface of TC and the surface of IC so that the PD-L1 bound to PD-1 on the surface of IC may induce adaptive immune resistance, leading to poor survival, while the PD-L1 remaining on the surface of TC are not. Previous research has revealed that in addition to tissue PD-L1, there was also existing circulating PD-L1 such as exosomal PD-L1 [52,53] and soluble PD-L1 [54,55]. However, further investigations are essential to verify the mechanisms behind this hypothesis.

In summary, this meta-analysis demonstrated a significant association of PD-L1 expression with advance stage (both TC and IC) and lympho-vascular invasion (only IC). The PD-1/PD-L1 signalling pathway in cancer has been explored actively since the approval of pembrolizumab for the treatment of melanoma in September 2014 [56,57,58]. PD-L1 works as a pro-tumourigenic factor in cancer cells by connecting to its receptors and activating proliferation and survival signalling pathways [59,60]. This discovery strengthens the evidence of the involvement of PD-L1 in tumour progression. Furthermore, PD-L1 has been reported to have non-immune proliferative effects on tumour cell types [59]. Hence, changes in molecular pathogenesis including immune checkpoint components need to be understood to improve the treatment regimen for endometrial cancer patients. Current EC management is by surgical intervention due to its recurring and metastasis behaviour, and women with EC often have poorer prognosis in the advance stage [17,61]. These meta-analysis results have significantly upheld the association of PD-L1 expression to the occurrence of EC.

The expression of PD-L1 from both TC and IC could predict the sensitivity to PD-1/PD-L1 axis targeted therapeutics. Recently, immunotherapy has emerged as a new option for endometrial cancer treatment. Studies have shown that therapies that break the PD-1 and PD-L1 interaction such as pembrolizumab, nivolumab, atezolizumab, and avelumab had an anti-tumour effect on melanoma, non-small cell lung carcinoma, renal cell cancer, and Hodgkin lymphoma. The preliminary results reported by the American Society of Clinical Oncology showed that atezolizumab and pembrolizumab might have promising outcomes in the treatment of endometrial cancer. To date, six clinical trials are in active ongoing phase III combination immunotherapy trials in patients with advanced stage and recurrent EC [62,63]. In addition, more data have demonstrated that cytotoxic drugs and targeted therapies influence immune responses [17,61]. Recent clinical studies in the immunotherapy of PD-1/PD-L1 showed positive significant responses for many advanced cancers such as gastric and gastro-oesophageal junction adenocarcinomas, non-small cell lung cancer, and malignant melanoma [64,65,66]. As PD-L1 has been used as a predictive value to prognosticate the disease progression, it can be used to guide the use of an immunotherapy inhibitor of PD-L1/PD-1. Current clinical trials of EC patients with combined treatment of pembrolizumab and lenvatinib showed promising results, with an overall response rate of 38% and median progression-free survival (PFS) of 7.4 months, and median OS of 16.7 months [63,67,68]. Hence, we postulate that this value/clinical application may help clinicians to utilize this anti-checkpoint inhibitor as an adjuvant therapy to improve survival outcome.

Despite a comprehensive meta-analysis, numerous limitations should be considered when interpreting our findings. First, the cut-off points of the age group were not standardized among the included studies. It would be of great benefit to ensure that a complete dataset of age is provided for more accurate analysis with the age group. Second, the eligible studies used different antibodies to determine the PD-L1 positivity. These variances in multiple methodologies could lead to potential heterogeneity. Third, the method of analysis should widely use qPCR for diagnosis and analysis for better validation and comparison. Finally, the data for survival analysis in EC with PD-L1 expression were limited as the individual study focused on TC and IC separately and only two studies included in the meta-analysis led to heterogeneity. Thus, it is recommended that the details of each study be standardized, which is essential for further analysis and can later be translated into significant outcomes.

## 5. Conclusions

In conclusion, this meta-analysis provided an overview of the prevalence of PD-L1 expression in EC and its association with clinicopathological features. PD-L1 expression was significantly associated with advanced tumour stage (Stage III/IV) in TC and IC and associated with the presence of lympho-vascular invasion in IC. However, PD-L1 expression in TC was not associated with age group, histology type, myometrial invasion, and lympho-vascular invasion. In IC, PD-L1 expression was not associated with age group, histology type, and myometrial invasion. Furthermore, the meta-analysis of survival outcomes of a high expression of PD-L1 had a significant association with worse OS in IC but not in TC. Clinically, these findings could be utilized to improve the therapeutic modalities and find a possible new biomarker for the early diagnosis of EC. This information may provide an evidence-based medicine for clinicians to stratify EC patients for anti-PD-1/PD-L1 immune therapy, especially patients with high PD-L1 expression in IC with advanced cancer or lympho-vascular invasion. However, well-designed cohort studies are needed to verify the association of PD-L1 expression in EC before applying this promising targeted therapy.

## Figures and Tables

**Figure 1 cancers-14-03911-f001:**
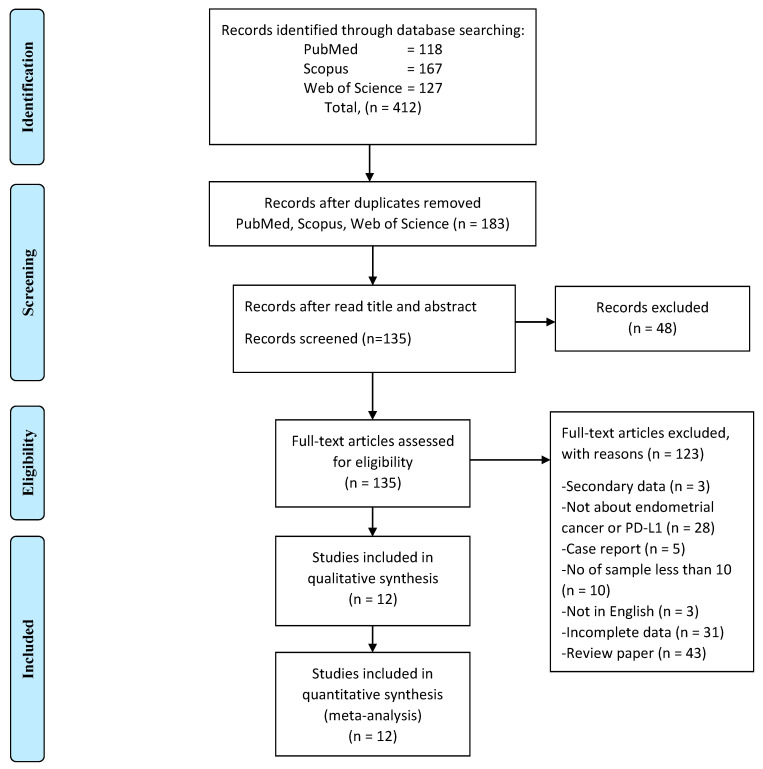
The PRISMA flow diagram of study selection and screening.

**Figure 2 cancers-14-03911-f002:**
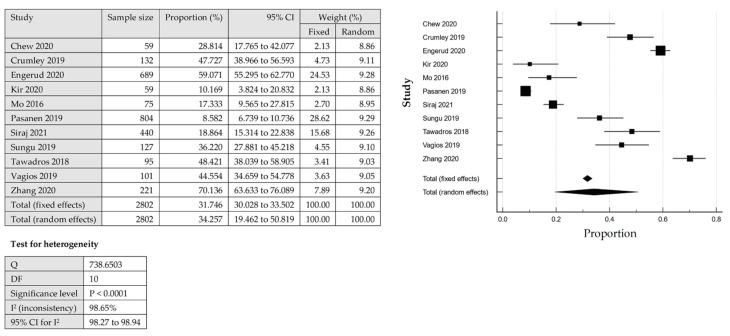
The pooled prevalence of PD-LI positive expression in the tumour cells (TCs) among endometrial cancer cases [17,18,19,24,25,26,27,28,29,30,31].

**Figure 3 cancers-14-03911-f003:**
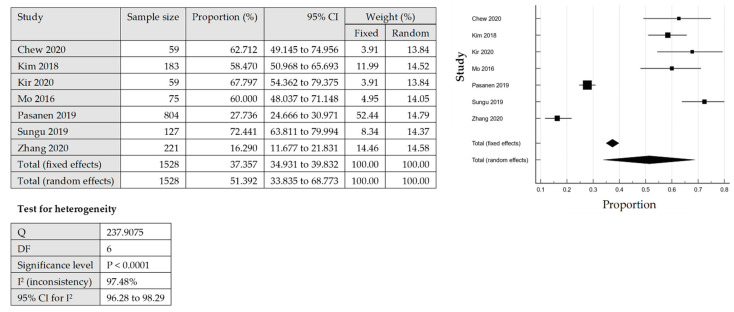
The pooled prevalence of PD-LI positive expression in immune cells (ICs) among the endometrial cancer cases [19,24,26,27,28,31,32].

**Figure 4 cancers-14-03911-f004:**
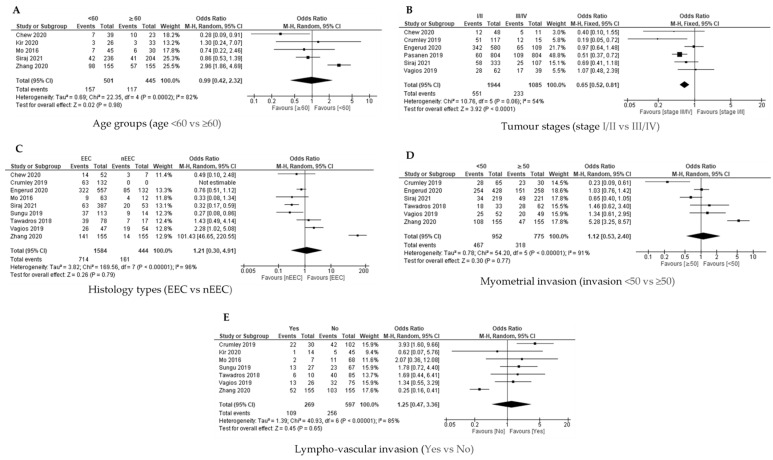
The PD-L1 positive expression among the endometrial cancer cases in tumour cells (TCs) [17,18,19,24,25,26,27,28,29,30,31]. Forest plot of (**A**) age group, (**B**) tumour stage, (**C**) histological type, (**D**) myometrial invasion, (**E**) lympho-vascular invasion.

**Figure 5 cancers-14-03911-f005:**
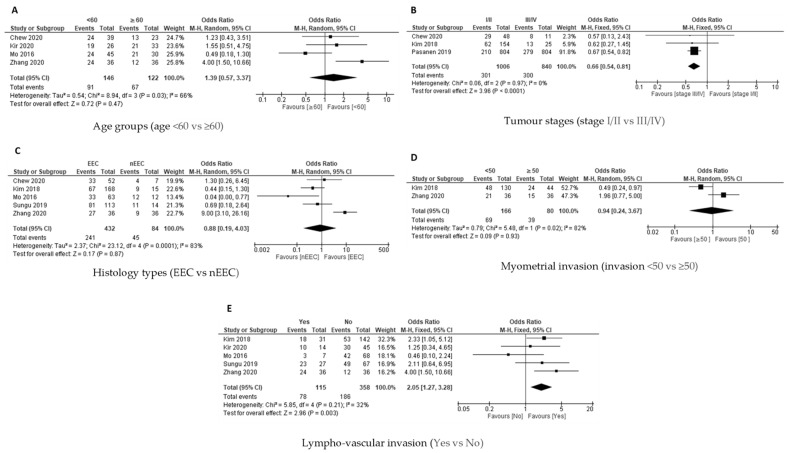
The PD-L1 positive expression among endometrial cancer cases in immune cells (ICs) [19,24,26,27,28,31,32]. Forest plot of (**A**) age group, (**B**) tumour stages, (**C**) histological types, (**D**) myometrial invasion, (**E**) lympho-vascular invasion.

**Figure 6 cancers-14-03911-f006:**
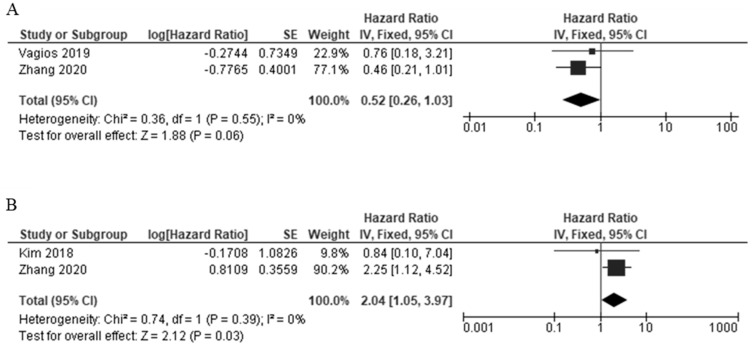
The forest plots of the studies evaluating the hazard ratio (HR) and 95% confidence interval (CI) of high PD-L1 expression in EC [17,24,32]. (**A**) Forest plots of overall survival (OS) in TCs. (**B**) Forest plots of overall survival (OS) in ICs.

**Figure 7 cancers-14-03911-f007:**
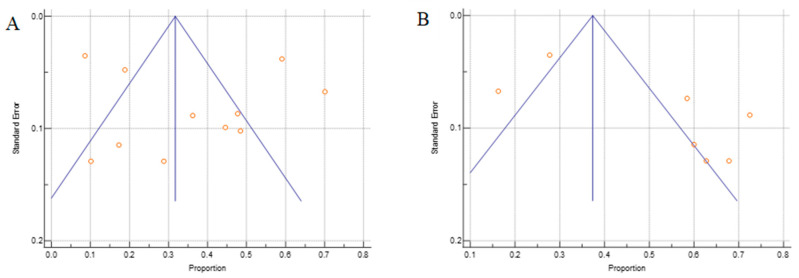
The Begg’s funnel plot of the PD-L1 proportion for the (**A**) TCs and (**B**) ICs. Side blue line indicates 95% confidence interval, middle blue line is the overall effect, and orange dots represent each individual study.

**Figure 8 cancers-14-03911-f008:**
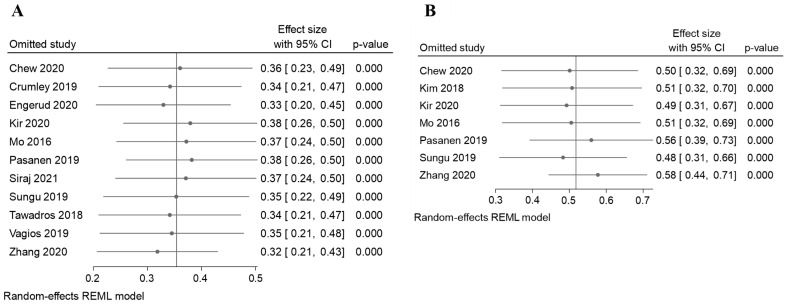
The sensitivity analysis for the results of the meta-analysis [17,18,19,24,25,26,27,28,29,30,31,32]. The sensitivity analysis to evaluate the stability of the pooled results for the PD-L1 positive proportion of EC cases in the (**A**) TCs (**B**) ICs.

**Table 1 cancers-14-03911-t001:** The risk of bias (NOS cohort).

Authors	Domains			Results	
	Selection	Comparability	Outcome	Score	Risk
Chew 2020 [31]	****	**	**	8	Low
Crumley 2019 [30]	****	*	**	7	Low
Engerud 2020 [29]	****	**	**	8	Low
Kim 2018 [32]	****	*	*	6	Low
Kir 2020 [28]	****	*	**	7	Low
Mo 2016 [19]	****	**	**	8	Low
Pasanen 2019 [27]	****	**	**	8	Low
Siraj 2021 [18]	****	*	**	7	Low
Sungu 2019 [26]	****	**	**	8	Low
Tawadros 2018 [25]	****	*	**	7	Low
Vagios 2019 [17]	****	*	**	7	Low
Zhang 2020 [24]	****	*	**	7	Low

The NOS assessment contains nine stars divided into three categories: selection, comparability and outcome. Each star will be given score of 1 (**** = score of 4; ** = score of 2; * = score of 1). Score of ≥6 indicates the study is low risk of bias.

**Table 2 cancers-14-03911-t002:** A summary of 12 studies that included cases of PD-L1 positive expression with clinicopathological factors among patients with endometrial cancer.

Author/Year	Country	Study Design	No. of Sample	No. of PD-L1+ (%)		Age of Diagnosis	Tumour Stage	Histology Type	Myometrial Invasion	Lympho- Vascular Invasion
Chew 2020 [31]	Malaysia	Retrospective	59	TC	28.8	<60 = 7/39	I/II = 12/48	EEC = 14/52	NA	NA
						≥60 = 10/23	III/IV = 5/11	nEEC = 3/7		
				IC	62.7	<60 = 24/39	I/II = 29/48	EEC = 33/52	NA	NA
						≥60 = 13/23	III/IV = 8/11	nEEC = 4/7		
Crumley 2019 [30]	USA	Retrospective	132	TC	47.7	NA	I/II = 51/117	EEC = 63/132	<50 = 28/65	yes = 22/30
							III/IV = 12/15		≥50 = 23/30	no = 41/102
Engerud 2020 [29]	Norway	Prospective	689	TC	59.1	<66 = 203/350	I/II = 342/580	EEC = 322/557	<50 = 254/428	NA
						≥66 = 204/339	III/IV = 65/109	nEEC = 85/132	≥50 = 151/258	
Kim 2018 [32]	Korea	Retrospective	183	IC	58.6	≤55 = 42/110	I/II = 62/154	EEC = 67/168	<50 = 48/130	yes = 18/31
						>55 = 34/73	III/IV = 13/25	nEEC = 9/15	≥50 = 24/44	no = 53/142
Kir 2020 [28]	Turkey	Retrospective	59	TC	10.2	<60 = 3/26	NA	NA	NA	yes = 1/14
						≥60 = 3/33				no = 5/45
				IC	67.8	<60 = 19/26	NA	NA	NA	yes = 10/14
						≥60 = 21/33				no = 30/45
Mo 2016 [19]	USA	Retrospective	75	TC	17.3	<60 = 7/45	NA	EEC = 9/63	NA	yes = 2/7
						≥60 = 6/30		nEEC = 4/12		no = 11/68
				IC	60.0	<60 = 24/45	NA	EEC = 33/63	NA	yes = 3/7
						≥60 = 21/30		nEEC = 12/12		no = 42/68
Pasanen 2019 [27]	Finland	Retrospective	842	TC	8.6	NA	I/II = 60/804	NA	NA	NA
							III/IV = 109/804			
				IC	27.7	NA	I/II = 210/804	NA	NA	NA
							III/IV = 279/804			
Siraj 2021 [18]	Saudi Arabia	Retrospective	440	TC	18.9	<60 = 42/236	I/II = 58/333	EEC = 63/387	<50 = 34/219	NA
						≥60 = 41/204	III/IV = 25/107	nEEC = 20/53	≥50 = 49/221	
Sungu 2019 [26]	Turkey	Retrospective	127	TC	36.2	NA	NA	EEC = 37/113	NA	yes = 13/27
								nEEC = 9/14		no = 23/67
				IC	72.4	NA	NA	EEC = 81/113	NA	yes = 23/27
								nEEC = 11/14		no = 49/67
Tawadros 2018 [25]	Egypt	Retrospective	95	TC	48.4	<50 = 17/41	NA	EEC = 39/78	≥50 = 28/62	yes = 6/10
						≥50 = 29/54		nEEC = 7/17	<50 = 18/33	no = 40/85
Vagios 2019 [17]	Greece	Retrospective	101	TC	44.6	NA	I/II = 28/62	EEC = 26/47	≥50 = 20/49	yes = 13/26
							III/IV = 17/39	nEEC = 19/54	<50 = 25/52	no = 32/75
Zhang 2020 [24]	Japan	Retrospective	221	TC	70.1	<60 = 98/155	NA	EEC = 141/155	≥50 = 47/155	yes = 52/155
						≥60 = 57/155		nEEC = 14/155	<50 = 108/155	no = 103/155
				IC	16.3	<60 = 24/36	NA	EEC = 27/36	≥50 = 21/36	yes = 24/36
						≥60 = 12/36		nEEC = 9/36	<50 = 15/36	no = 12/36

TC = tumour cell; IC = immune cell; EEC = endometroid endometrial cancer; nEEC = non-endometroid endometrial cancer; NA = data not available.

**Table 3 cancers-14-03911-t003:** The robustness check of the meta-analysis results using the leave-one-out method.

Analysis of PD-L1 Positive Expression	Overall ES	Leave-One-Out Result			
		Lowest Study Range	*p*-Value	Highest Study Range	*p*-Value
*Proportion*					
TC	0.34	0.20–0.45	0.000	0.26–0.50	0.000
IC	0.52	0.31–0.66	0.000	0.39–0.73	0.000
*Clinicopathological factor and TC*					
Age group	0.99	0.34–1.07	0.000	0.58–1.72	0.000
Tumour stage	0.65	0.44–0.75	0.000	0.60–1.03	0.000
Histology type	0.70	0.30–0.73	0.000	0.59–1.06	0.000
Myometrial invasion	1.02	0.61–1.08	0.000	0.63–1.65	0.000
Lympho-vascular invasion	0.76	0.36–0.76	0.000	0.76–1.97	0.000
*Clinicopathological factor and IC*					
Age group	1.39	0.42–1.45	0.000	0.56–1.92	0.000
Tumour stage	0.75	0.60–0.90	0.000	0.26–1.31	0.003
Histology type	0.69	0.24–1.05	0.002	0.34–1.33	0.001
Myometrial invasion	0.92	0.25–1.11	0.002	0.21–2.59	0.021
Lympho-vascular invasion	1.22	0.56–1.70	0.000	0.76–1.86	0.000
*Survival analysis, OS*					
TC	0.52	0.06–0.86	0.024	−0.76–2.28	0.326
IC	2.04	−2.63–4.31	0.635	0.55–3.95	0.009

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
