# Peer review of "PD-L1 Expression in Endometrial Cancer and Its Association with Clinicopathological Features: A Systematic Review and Meta-Analysis"

_cancers, 2022, doi:10.3390/cancers14163911_

Round 1

Reviewer 1 Report

Tables shown in Figures 2a and 3a are unnecessary to be shown, excess data could be mentioned in the text.

Author Response

We appreciated the suggestion, and we combined Figure 2(a,b) and 3(a,b) into just Figure 2 and Figure 3 to make it less confusing. As the information is essential to represent the weight of individual studies for analysis (as also included in Figures 4 & 5), we would like to maintain these figures and summarize the information in text.

Reviewer 2 Report

The authors have improved the manuscript based on previous comments. However, the authors can expand the conclusion to highlight the clinical relevance.

Author Response

The conclusion had been expended as suggested and was highlighted in yellow colour. 

Round 2

Reviewer 2 Report

The Authors have improved the manuscript based on the comments. 

This manuscript is a resubmission of an earlier submission. The following is a list of the peer review reports and author responses from that submission.

Round 1

Reviewer 1 Report

In this manuscript entitled “PD-L1 expression……..”, Yusof et al. conducted a systematic review and a metanalysis focusing on the role of PDL-1 expression in prognosis of EC. Although authors have tried to explain that the association of PDL-1 expression with EC is relatively new, however there are many literatures citing this association over a decade and this area of research is rapidly growing.

Overall there are lots of concerns in this manuscript as the meta analysis is not appropriate and sometime more confusing. The manuscript is not written in coherent fashion therefore there is more confusion than a stepwise clear analysis. The method is not written clearly and the end result is not coherent as well.

I would suggest the authors to do more coherent analysis, highlight the importance of the study and the need of a robust analysis with elaborated method to make the central point of the paper more accurate. Extensive re-hauling of the manuscript is to be done to make it fit for the consideration. Therefore I am compelled to reject this manuscript in this current form.

Providing some examples of the flaws in the manuscript….

  • Line 17-18 : “The recent finding…………treatment” The finding of PDL-1 expression is not recent in EC, also the following statement “has been explored previously” Kindly modify the statement to make it look subtle and appropriate
  • Line 47-51 : “Cancer oncogenesis…….immune system”. Kindly modify the statement and make it more precise by highlighting the importance. The statement is in favor of cancer cells and not in favor of immune system. Kindly re-phrase or remove this part.

There is a series of confusion in this manuscript related to qualitative and well as quantitave analysis.

Reviewer 2 Report

Abstract: Includes large section of introduction with little part describing in depth the results and conclusion.

Introduction:

Line 38: updated statistics for 2022 could be retrieved from SEER database online.

Line 40, extra spaces.

Methodology:

What is the definition of positive expression? at particular cutoff? e.g 10%, median. Some articles rely on scoring of positive at >5% or 10%, .... So was it consistent across the articles. Please add the definition to the methodology. Line 132 showed limitation related to this concern, but the variation across the studies were not explained anywhere.

Results: significant heterogeneity was found, meta-regression, subgroup analysis, and sensitivity analysis are required to identify the reason of such heterogeneity.

Publication bias was not performed.

Association of PDL1 with survival could be retrieved from online databases as another validation

e.g some examples (but there are more sources)

https://cptac-data-portal.georgetown.edu/

CPTAC (Clinical Proteomic Tumor Analysis Consortium)

https://www.proteinatlas.org/

http://kmplot.com/

Reviewer 3 Report

Journal

Cancers (ISSN 2072-6694)

Manuscript ID

cancers-1707682

Type

Systematic Review

Title

PD-L1 expression in endometrial cancer and its association with clinicopathological features: a systematic review and meta-analysis

Authors

Mohd Nazzary Mamat @ Yusof , Kah Teik Chew * , Nirmala Kampan , Norhaslinda Abdul Aziz , Reena Rahayu Md Zin , Geok Chin Tan , Mohamad Nasir Shafiee

Section

Systematic Review or Meta-Analysis in Cancer Research

In this manuscript, Yusof et al have reviewed the role of PDL1 in endometrial cancer in the form of a meta-analysis. This is an interesting manuscript with some scope for improvement. Authors need to discuss this finding in the context of the clinical application, the relevance of immunotherapy, and immune-related adverse events. These additional points of discussion will add significant strength to the manuscript.

Please find my other comments below:

  1. The authors can add a sentence to the abstract so that its beginning does not seem abstract.  Refer 'At present, treatment for endometrial cancer (EC) is mainly surgical removal of the tu- 13 mour bulk; adjuvant radiotherapy with or without chemotherapy/hormonal therapy are applied. '
  2. Authors can add a brief introduction to Endometrial cancer in the introduction. Refer 'Endometrial cancer (EC) is a heterogeneous group of tumors derived from endo- 35 metrial glandular epithelium 1,2. According to Global Cancer Statistics 2020, EC was the 36 6th most common diagnosed cancer among women world widely.'
  3. Authors need to expand this section "  PD-L1 has been studied for its predictive value in EC with mixed results 16–18. 69 Several studies had found that PD-L1 expression does not have a substantial prognostic 70 effect in EC 16."
  4. Authors can add a discussion on immunotherapy. 
  5. Follow Oaknin, Ana, Alicia León-Castillo, and Domenica Lorusso. "Progress in the management of endometrial cancer (subtypes, immunotherapy, alterations in PIK3CA pathway): data and perspectives." Current Opinion in Oncology 32, no. 5 (2020): 471-480.
  1. Other reference that can enrich the discussion,  "Makker, V., Taylor, M.H., Aghajanian, C., Oaknin, A., Mier, J., Cohn, A.L., Romeo, M., Bratos, R., Brose, M.S., DiSimone, C. and Messing, M., 2020. Lenvatinib plus pembrolizumab in patients with advanced endometrial cancer. Journal of clinical oncology, 38(26), p.2981." and "Makker, Vicky, Drew Rasco, Nicholas J. Vogelzang, Marcia S. Brose, Allen L. Cohn, James Mier, Christopher Di Simone et al. "Lenvatinib plus pembrolizumab in patients with advanced endometrial cancer: an interim analysis of a multicentre, open-label, single-arm, phase 2 trial." The Lancet Oncology 20, no. 5 (2019): 711-718."
  2. The authors needs to discuss immune-related adverse events (irAE) Refer "Halla, Kimberly. "Emerging Treatment Options for Advanced or Recurrent Endometrial Cancer." Journal of the Advanced Practitioner in Oncology 13, no. 1 (2022): 45." and "Jing, Y., Liu, J., Ye, Y., Pan, L., Deng, H., Wang, Y., Yang, Y., Diao, L., Lin, S.H., Mills, G.B. and Zhuang, G., 2020. Multi-omics prediction of immune-related adverse events during checkpoint immunotherapy. Nature communications, 11(1), pp.1-7."